# An Elucidation of the Anti-Photoaging Efficacy and Molecular Mechanisms of Epigallocatechin Gallate Nanoparticles in a Balb/c Murine Model

**DOI:** 10.3390/foods14132150

**Published:** 2025-06-20

**Authors:** Fangzhi Xia, Fei Wang, Liangchun Kuo, Pengyu Huang, Anyi Liu, Gao Wang, Xiaomin Tang, Kun Guan, Ying Xie, Junbo Wang

**Affiliations:** 1Department of Nutrition and Food Hygiene, School of Public Health, Peking University, Beijing 100191, China; 2211210080@stu.pku.edu.cn (F.X.); wangfei.duzhuo@163.com (F.W.); 2211210081@bjmu.edu.cn (L.K.); 2211120010@bjmu.edu.cn (A.L.); 2010306103@bjmu.edu.cn (G.W.); tangxiaomir1618@163.com (X.T.); guankuns@163.com (K.G.); 2State Key Laboratory of Natural and Biomimetic Drugs, Beijing Key Laboratory of Molecular Pharmaceutics and Drug Delivery System, and School of Pharmaceutical Sciences, Peking University, Beijing 100191, China; 2311210059@bjmu.edu.cn (P.H.); bmuxieying@bjmu.edu.cn (Y.X.); 3School of Public Health, Peking University, and Beijing Key Laboratory of Toxicological Research and Risk Assessment for Food Safety, Beijing 100191, China

**Keywords:** EGCG, skin aging, ultraviolet rays, nanoparticle

## Abstract

With the increasing frequency of ultraviolet (UV) exposure in daily life and the exploration of anti-photoaging strategies, natural plant-derived compounds with anti-skin-aging properties have garnered significant attention. This study aimed to evaluate the efficacy of zein-chitosan-based nanocarriers in enhancing the bioavailability of epigallocatechin gallate (EGCG) and to elucidate its mechanisms in ameliorating skin photoaging. Utilizing a Balb/c mouse model of photoaging, we monitored skin conditions, analyzed skin barrier function parameters, and observed changes in skin tissue structure and collagen fibers through hematoxylin–eosin (H&E) and Masson staining. Immunohistochemical staining was employed to assess COL1A1 levels in the skin, while enzyme-linked immunosorbent assay (ELISA) was used to measure antioxidant enzymes, inflammatory cytokines, matrix metalloproteinases (MMPs), and NF-kB levels. The effects of orally administered EGCG nanoparticles on UV-induced skin aging were investigated. UV exposure significantly increased skin roughness, impaired skin barrier function, thickened the epidermis, reduced collagen content, decreased antioxidant enzyme activity, and elevated levels of inflammatory cytokines, MMPs, and NF-kB in the model group compared to the normal control group. EGCG nanoparticles markedly ameliorated these photoaging manifestations, with some indicators showing superior improvement compared to free EGCG. These findings suggest that EGCG nanoparticles exhibit enhanced anti-photoaging effects over free EGCG, highlighting the potential of nanocarriers as a promising strategy to improve the bioavailability of EGCG.

## 1. Introduction

The skin, the largest organ of the human body, comprises three layers: the epidermis, dermis, and subcutaneous tissue. As the interface between the internal and external environments, the skin plays a crucial role in thermoregulation, fluid retention, protection against external stressors, and the modulation of tactile and nociceptive sensations [1,2]. Ultraviolet (UV) radiation is the primary exogenous factor (accounting for over 80%) accelerating skin aging and hastening the natural aging process [3]. Extensive epidemiological studies have demonstrated that the development of various dermatological conditions, such as actinic elastosis, actinic keratosis, and chronic actinic cheilitis, is directly or indirectly linked to skin photoaging. In severe cases, it can even lead to basal cell carcinoma, squamous cell carcinoma, and melanoma [4]. Consequently, the issue of skin photoaging has garnered increasing attention.

Current anti-photoaging strategies are broadly categorized into physical interventions and chemical treatments, each with inherent limitations. Physical therapies for photoaging primarily include laser techniques, high-intensity focused ultrasound (HIFU), and dermal filler injections, while severe cases may require surgical intervention [5,6]. For instance, However, any standalone physical modality often fails to achieve satisfactory therapeutic outcomes. Moreover, surgical procedures are inherently invasive, associated with prolonged recovery periods and potential risks. Beyond physical approaches, the application of sunscreen products to mitigate UV radiation remains a widely adopted strategy. Nevertheless, sunscreens require frequent reapplication to maintain efficacy, fail to reverse pre-existing molecular damage [7], and often contain chemical additives (e.g., titanium dioxide) that may exert toxic effects and pose health risks upon prolonged exposure [8]. Pharmacological agents such as estrogen (E2) and glucocorticoids have also been employed against photoaging, yet their utility is limited by hormone-induced irritancy, dependency, and potential endocrine disruption with chronic use [9]. Vitamins represent another common anti-photoaging intervention. For instance, vitamin A derivatives attenuate fine wrinkles, hyperpigmentation, and skin laxity by suppressing inflammatory cytokines and downregulating matrix metalloproteinase (MMP) activity [10]. However, improper use may provoke adverse effects, including erythema, desquamation, xerosis, and pruritus [11]. Similarly, vitamin C—despite its widespread use—is hampered by poor chemical stability and inadequate skin permeability, significantly restricting its clinical efficacy [12]. Therefore, the development of highly effective and low-side-effect therapeutic approaches has become a focal point of research. Natural plant-derived compounds and herbal formulations have gained popularity due to their rich biological activities and safety profiles.

Epigallocatechin gallate (EGCG), the most biologically active and abundant catechin, constitutes 50% to 70% of total catechins [13]. With a molecular weight of 458.38 Da and the molecular formula C22H18O11, EGCG is a potent antioxidant [14]. Its unique structure, featuring multiple hydroxyl, amine, and benzene groups, endows EGCG with inherent UV-absorbing capabilities [15,16]. However, its polyhydroxy structure also results in low intestinal permeability, extreme susceptibility to oxidation in alkaline environments, poor stability in the gastrointestinal tract, a short half-life, and a rapid first-pass metabolism, significantly reducing its bioavailability in vivo [17,18].

Recent studies have shown that nanostructures can enhance EGCG stability across different pH levels, prolong its intestinal retention time, and substantially improve its bioavailability [19,20]. Nevertheless, the metabolic mechanisms and long-term safety of nanocarriers as a means to enhance EGCG bioavailability remain unclear. Moreover, current research on EGCG nanoparticles predominantly focuses on their antioxidant effects, with limited detailed analysis of their mechanisms in combating skin photoaging. This study aims to evaluate the extent to which nanostructures enhance EGCG bioavailability and to explore the anti-photoaging effects and mechanisms of EGCG nanoparticles.

## 2. Materials and Methods

### 2.1. Preparation of Epigallocatechin Gallate Nanoparticles (EGCG-NPs)

#### 2.1.1. Preparation of Blank Nanoparticles

An 80% (*v*/*v*) ethanol–water solution (pH 4, adjusted with 0.10 mol/L HCl) was prepared. Zein (100.0 mg) was dissolved in this solution to yield a 10.0 mg/mL zein solution. Chitosan (CTS, 20.0 mg) was dissolved in a 1% (*v*/*v*) acetic acid–water solution to prepare a 0.67 mg/mL CTS solution. The zein solution was added dropwise into the CTS solution under magnetic stirring (200 rpm). The organic solvent was removed by rotary evaporation at 40 °C, and the solution was diluted to 10.0 mL with ultrapure water to obtain a bluish opalescent blank nanoparticle solution, which was stored in the dark at 4 °C.

#### 2.1.2. Preparation of EGCG-NPs

According to a mass ratio of Zein:EGCG = 10:1, 10.0 mg of EGCG powder and 100.0 mg of Zein powder were co-dissolved in an 80% (*v*/*v*) ethanol–water solution. The subsequent steps were performed following the same procedure as described for the preparation of blank nanoparticles.

### 2.2. Animal Care and Skin Aging Model

Seventy-five 6-week-old male SPF (specific pathogen free) Balb/c mice (provided by the Department of Laboratory Animal Science, Peking University Health Science Center), weighing 20 ± 2 g, were randomly divided into five groups (n = 15 per group): normal control, UV model, blank nanoparticle model, EGCG treatment, and EGCG nanoparticle treatment. We determined the sample size based on previous literature in the field while ensuring sufficient statistical power for reliable conclusions upon study completion. No explicit exclusion criteria were established in this experiment, and all animal data were included in the final statistical analysis. A fixed photoperiod was implemented consistently, with the daily randomization of illumination sequences among cages, and all cage locations were kept stable during the experimental period. All mice were housed in an SPF-grade animal facility at the Department of Laboratory Animal Science, Peking University Health Science Center, with ad libitum access to food and water. The facility maintained an ambient temperature of 24–26 °C, a relative humidity of 50–60%, and a 12 h:12 h light–dark cycle. The animal experimental protocol was approved by the Ethics Committee of Peking University Health Science Center (Approval No.: LA2022393). All animal care, management, and experimental procedures were conducted in accordance with the “Beijing Municipal Regulations on Laboratory Animals.”

The dorsal areas of all the mice were shaved. A UV irradiation source was constructed by alternating two 40 W UVA tubes (peak wavelength: 340 nm) and two 40 W UVB tubes (peak wavelength: 313 nm) connected in parallel, positioned 40 cm above the mice. Preliminary experiments determined the minimal erythema dose (MED) to be 20 min. All mice were acclimatized for one week prior to experimentation until their physiological and behavioral status stabilized. To better simulate the in vivo metabolic patterns of EGCG and EGCG-NPs while avoiding mechanical skin damage in mice, we selected oral gavage as the administration method. The normal control group (Normal) received no treatment. The UV model control group (UV) was exposed to 1/4 MED daily. The blank nanoparticle model group (SPNP) received 1/4 MED daily and was orally administered 0.1 mL/10 g of blank nanoparticle solution. The EGCG treatment group (EGCG) received 1/4 MED daily and was orally administered 0.1 mL/10 g of EGCG aqueous solution (1.1 mg/g body weight). The EGCG nanoparticle group (DPNP) received 1/4 MED daily and was orally administered 0.11 mL/10 g of EGCG nanoparticle solution (1 mg/mL). The experiment lasted for 8 weeks. The general condition, behavior, and skin status of the mice in all groups were monitored daily. Body weight measurements were initiated on the first day of intervention and recorded weekly (on the final day of each week) from Week 1 to Week 8. If severe dorsal skin ulceration was observed in the mice during the experiment, UV irradiation would be discontinued until the lesions showed signs of recovery. Throughout the study, only one researcher (F.X.) was aware of the specific group allocations and experimental procedures. All groups received identical UV exposure durations, ensuring consistent treatment regardless of group assignment. The details of animal grouping and experimental treatments are presented in Table 1, and the overall experimental schedule is depicted in Figure 1.

### 2.3. Assessment of Skin Barrier Function

At three time points during the experiment (Week 1 [W1], Week 4 [W4], and Week 8 [W8]), skin barrier function was evaluated using the Dermalab Combo4 system (Cortex Technology, Aalborg, Denmark). The following parameters were measured on the shaved dorsal skin of the mice: skin hydration, skin elasticity, and ultrasonographic characteristics. Quantitative data and imaging analyses were performed to assess epidermal roughness and collagen density.

### 2.4. Tissue Harvesting and Organ Index Calculation

After 8 weeks of intervention, the mice were euthanized by exsanguination via enucleation. The dorsal skin from the UV-exposed area was excised, and a 1 cm × 1 cm section from the central region was fixed in 4% paraformaldehyde solution for at least 24 h. The remaining skin tissue was stored at −80 °C for further analysis.

Major organs, including the heart, liver, spleen, and kidneys, were harvested, weighed, and recorded. The organ index was calculated using the following formula (Equation (1)):Organ Index (mg/g) = Organ Weight (mg)/Fasting Body Weight (g)(1)

### 2.5. H&E Staining of Liver and Skin Tissues

Following the procedures described in Section 2.4, skin and organ samples fixed in 4% paraformaldehyde were processed for histological analysis.

Liver and skin tissues from each group were paraffin-embedded, sectioned, and stained with H&E for histopathological evaluation using an upright microscope. The paraffin sectioning thickness was set at 4 μm for skin tissues and 3 μm for liver tissues.

For skin tissues, Masson’s trichrome staining was performed to assess changes in collagen fiber morphology and collagen content under an upright microscope. Additionally, immunohistochemical (IHC) staining for collagen type I (COL1A1) was conducted on skin samples from three randomly selected mice per group. The distribution and expression levels of COL1A1 were analyzed using microscopic imaging and quantitative image analysis.

### 2.6. Determination of Oxidative Stress Markers in Skin Tissue

Skin tissues were homogenized into a fine powder using a high-throughput liquid nitrogen grinding system. A precisely weighed quantity of skin powder was mixed with PBS at a 1:9 (*w*/*v*) ratio, followed by centrifugation at 3000 rpm for 20 min at 4 °C to obtain the supernatant for subsequent analyses.

The levels of oxidative stress markers were quantitatively assessed using enzyme-linked immunosorbent assay (ELISA) according to the manufacturer’s protocols (Beijing Beilai Biotechnology Limited, Beijing, China). The following parameters were measured: antioxidant enzyme activities/superoxide dismutase (SOD), glutathione peroxidase (GSH-Px), catalase (CAT), and oxidative stress marker/malondialdehyde (MDA) concentration.

### 2.7. Determination of Inflammatory Markers in Skin Tissue

The preparation of skin homogenate for analysis followed the same procedure as described in Section 2.6.

The levels of inflammatory cytokine markers were quantitatively assessed using ELISA according to the manufacturer’s protocols (Beijing Beilai Biotechnology Limited, Beijing, China). The following parameters were measured: pro-inflammatory cytokines/Interleukin-6 (IL-6), Interleukin-1β (IL-1β).

### 2.8. Determination of Matrix Metalloproteinase (MMP) Markers in Skin Tissue

The preparation of skin homogenate for analysis followed the same procedure as described in Section 2.6.

The levels of matrix metalloproteinases markers were quantitatively assessed using ELISA according to the manufacturer’s protocols (Beijing Beilai Biotechnology Limited, Beijing, China). The following parameters were measured: matrix metalloproteinases MMP-1 and MMP-3.

### 2.9. Measurement of NF-κB Signaling Pathway Activity

Following blood collection via enucleation, whole blood samples were allowed to clot at room temperature for 30 min to facilitate serum separation. The samples were then centrifuged at 3000 rpm for 10 min at 4 °C to obtain serum. The serum was aliquoted into two 500 μL microcentrifuge tubes and stored at −80 °C until analysis.

The nuclear factor kappa B (NF-κB) levels in serum were quantitatively determined using ELISA according to the manufacturer’s protocol (Beijing Beilai Biotechnology Limited, Beijing, China).

### 2.10. Statistics

Data organization and graphical representation were performed using Microsoft Excel 2020. Statistical analyses were conducted using SPSS 25.0 software. Continuous variables were expressed as mean ± standard deviation ($\bar{x}$ ± SD). Intergroup differences were assessed using one-way analysis of variance (ANOVA) for multiple comparisons, followed by Student’s *t*-test for pairwise comparisons. *p* < 0.05 was considered statistically significant, while a *p* < 0.01 indicated highly significant differences.

## 3. Results

### 3.1. Effects of EGCG-NPs on General Condition and Body Weight Changes

Throughout the 8-week intervention period, the model group mice showed symptoms such as decreased mental state, slow movement, and susceptibility to stress. The treatment group mice showed significant improvements in movement, skin, and hair status compared to the model group. To evaluate the overall functional status of the mice during the experiment, the weights of the mice were measured weekly, and the weekly weight changes were recorded to compare whether there were differences in weight among the five groups of mice. During the experiment, the body weight of the mice in each group showed an overall upward trend, and there was no statistically significant difference in body weight between the groups (*p* > 0.05), indicating that the physical function of the mice in each group was good during the experiment and EGCG-NPs had no adverse effects on the health of the mice (Figure 2).

### 3.2. Effects of EGCG-NPs on Organ Indices and Hepatic Function

To evaluate the organ health status of mice, the heart, liver, spleen, and kidneys of mice in each group were weighed after the experiment to calculate the organ indices (Table 2), and the pathological structure of the liver was observed by H&E staining (Figure 3). Statistical analysis showed no significant differences in the main organ indices among the five groups (*p* > 0.05). The H&E staining results indicated that no significant abnormalities were detected in the histopathological examination of liver tissues in all groups. The liver tissue morphology was intact, with clear hepatic lobule structures, distinct hepatocyte boundaries, and orderly cell arrangement. The nuclei exhibited normal morphology, radiating outward from the central vein. Except for the UV group, which showed partial inflammatory cell infiltration, no other abnormalities were observed. The comprehensive results suggest that EGCG-NPs have no physiological toxicity to the liver and no adverse effects on the organs in mice.

### 3.3. Effects of EGCG-NPs on Recovery of Dorsal Skin Lesions

To evaluate the changes in the dorsal skin condition of mice, weekly photographic documentation of the dorsal skin was performed. Figure 4 illustrates the dorsal skin conditions of the Balb/c mice in each group at the first, fourth, and eighth weeks of the experiment. As the experiment progressed, the dorsal skin condition of the Normal group showed no significant changes, with no erythema or scaling observed. In the UV group, erythema appeared, with the erythema area gradually increasing, color intensifying, scaling becoming more pronounced, and wrinkles increasing. The SPNP group exhibited increased and deepened wrinkles, the appearance of scaling, and erythema, though the erythema color was lighter compared to the photoaging model group. The EGCG group showed mild erythema, noticeable wrinkles, and scaling, but the dorsal skin condition was superior to that of the UV and SPNP groups. The DPNP group demonstrated smoother and more delicate dorsal skin, with fine wrinkles and no scaling or erythema.

### 3.4. Effects of EGCG-NPs on Improvement of Skin Barrier Function

To assess skin barrier function, cutaneous hydration, elasticity, and collagen density were measured using a Dermalab Combo4 skin analyzer (Cortex Technology, Denmark). The figure of skin elasticity analysis (Figure 5) shows that UV irradiation significantly prolonged skin recoil time in the model groups compared to the normal controls (*p* < 0.01). Both the EGCG and EGCG-NP treatments markedly ameliorated UV-induced skin laxity, demonstrating significantly shorter recoil times than the UV group (*p* < 0.01). Notably, EGCG-NPs exhibited superior efficacy in improving skin elasticity compared to EGCG alone (*p* < 0.05). The EGCG group showed significantly higher skin hydration than the UV group (*p* < 0.05) (Figure 6A), though no statistical difference was observed compared to the SPNP group (*p* > 0.05). The DPNP group demonstrated significantly enhanced skin hydration compared to the UV, SPNP, and EGCG groups (*p* < 0.01). Both UV-exposed model groups exhibited significantly reduced collagen density versus normal controls (*p* < 0.05) (Figure 6B). While EGCG treatment increased collagen density compared to the UV group (*p* < 0.05), the DPNP group showed significantly greater collagen enhancement than the EGCG group (*p* < 0.05).

### 3.5. Effects of EGCG-NPs on Pathological Architecture of Skin and Collagen Content

The histopathological examination of murine skin morphology was conducted via H&E staining (Figure 7). In the Normal group, H&E staining of dorsal skin lesions revealed typical histological features, including a well-defined stratum corneum, epidermis, and dermis. The epidermal layer exhibited a thin and structurally intact architecture. In contrast, both the UV and SPNP groups demonstrated significant epidermal hyperplasia, disorganized dermal collagen fiber arrangement, partial fiber fragmentation, and prominent inflammatory cell infiltration. The EGCG and DPNP groups exhibited attenuated skin lesions compared to the model groups, with the DPNP group showing improved collagen fiber alignment and reduced fiber fragmentation. The epidermal thickness measurements obtained by instrumental analysis were consistent with the H&E staining results.

The Masson staining results are presented in Figure 8. Histological analysis revealed that in the Normal group, the collagen fibers in the dermal layer of mouse skin exhibited a loose and well-organized arrangement with normal morphological characteristics. In contrast, the Model group demonstrated a significant reduction in dermal collagen fiber content, accompanied by a disorganized arrangement of collagen bundles. Numerous collagen fibers appeared thickened, degenerated, and fragmented. Both the EGCG-treated and DPNP-treated groups showed increased dermal collagen fiber contents compared to the Model group. Notably, the DPNP group exhibited a more orderly and aligned collagen fiber organization.

The immunohistochemical staining results for COL1A1 in the skin tissues of each group are presented in Figure 9. In the Normal group, extensive positive staining areas were observed throughout the skin tissue. Both the UV-exposed group and the SPNP-treated group exhibited significantly reduced positive staining areas compared to the Normal group. The EGCG-treated and DPNP-treated groups demonstrated COL1A1 staining patterns similar to the Normal group, with positive staining areas significantly larger than those in the UV-exposed group (*p* < 0.05).

### 3.6. Effects of EGCG-NPs on Oxidative Stress Function of Skin

Studies have demonstrated that skin photoaging is primarily attributed to excessive reactive oxygen species (ROS) accumulation in the skin induced by ultraviolet (UV) radiation [21]. Endogenous antioxidant enzymes in skin cells, including superoxide dismutase (SOD), catalase (CAT), and glutathione peroxidase (GSH-Px), play a critical role in scavenging ROS and maintaining cellular redox homeostasis [22]. The results revealed that, compared to the Normal group, UV irradiation significantly reduced the levels of antioxidant enzymes and increased malondialdehyde (MDA) content in both the UV and SPNP groups (*p* < 0.05). In contrast, treatment with EGCG and EGCG-NPs significantly ameliorated the UV-induced reduction in antioxidant enzyme levels and the increase in the oxidative product MDA (*p* < 0.05). Notably, GSH-Px levels were markedly elevated in the EGCG-NP group (*p* < 0.01), and the DPNP group exhibited superior improvement compared to the EGCG group (*p* < 0.05). These findings suggest that EGCG-NPs more effectively suppress oxidative stress responses and mitigate oxidative-stress-induced damage (Figure 10).

### 3.7. Effects of EGCG-NPs on Anti-Inflammatory Function of Skin

The ELISA results for inflammatory cytokines IL-6 and IL-1β in the skin tissues of each group are presented in Figure 11. The data revealed that, compared to the Normal group, both the UV group and the SPNP group exhibited elevated expression levels of IL-6 and IL-1β. Notably, the increase in IL-6 was statistically significant (*p* < 0.01). In comparison to the model groups, treatment with EGCG and DPNPs significantly reduced IL-1β expression (*p* < 0.05). However, while IL-6 levels in the EGCG group showed no statistically significant difference, a marked reduction in IL-6 expression was observed in the DPNP group (*p* < 0.01).

### 3.8. Effects of EGCG-NPs on Expression Level of MMPs in Skin

The ELISA results of the MMP-1 and MMP-3 levels in mouse skin tissues across the experimental groups are presented in Figure 12. The data demonstrate that, compared with the Normal group, both the UV and SPNP groups exhibit significantly elevated levels of MMP-1 and MMP-3 (*p* < 0.05). In contrast, the EGCG and DPNP groups showed a marked reduction in MMP-1 and MMP-3 expression compared to the Model group, with statistical significance (*p* < 0.05). Notably, the DPNP group exhibited a more pronounced decrease in MMP-1 levels than the EGCG group (*p* < 0.05), along with a highly significant downward trend relative to the Model group (*p* < 0.01).

### 3.9. Effects of EGCG-NPs on NF-κB Signal Transduction Pathway

Exposure to UV radiation increases the levels of free radicals in the body, which can activate the NF-κB heterodimer composed of p65 and p50, thereby triggering inflammatory and immune responses [23]. As shown in Figure 13, in this study, compared to the Normal group, the expression levels of NF-κB were significantly elevated in both the UV and SPNP groups (*p* < 0.05). In contrast, the EGCG and DPNP groups exhibited reduced NF-κB expression levels compared to the two model groups, with statistically significant differences (*p* < 0.05). Notably, the DPNP group demonstrated a more pronounced reduction in NF-κB expression than the EGCG group (*p* < 0.05), showing a significant downward trend compared to the model group (*p* < 0.01).

## 4. Discussion

EGCG, as a potent antioxidant, exhibits significant preventive and therapeutic effects on skin photoaging. Studies have demonstrated that nanostructures can enhance the in vivo stability of EGCG, prolong its intestinal retention time, and overcome its structural and physicochemical limitations, thereby amplifying its antioxidant capacity and improving its bioavailability [20,24,25]. The manifestations of skin aging include epidermal thinning, reduced elasticity, increased wrinkles, enhanced scaling, and impaired skin barrier function [26]. In this study, observations of the dorsal skin in mice revealed that, compared to the model control group, both the EGCG group and the DPNP group exhibited significant improvements in photoaging symptoms such as skin laxity, erythema, wrinkles, and scaling. Notably, the DPNP group demonstrated superior ameliorative effects. In non-invasive assessments of skin barrier function, the DPNP group exhibited more pronounced improvements than the EGCG group in mitigating photoaging-induced reductions in skin hydration and elasticity. Furthermore, ultrasonic evaluation via the Dermalab Combo4 system indicated that the DPNP group achieved a greater increase in collagen density compared to the EGCG group.

UV-induced alterations in skin appearance stem from microenvironmental changes triggered by ultraviolet radiation, which initiate a cascade of responses including DNA damage, oxidative stress, and inflammatory reactions. These processes ultimately disrupt the structural integrity of the epidermis and dermis, manifesting as visible signs of photoaging. At the histopathological level, aged skin exhibits acanthosis (epidermal thickening), reduced dermal papillae, the flattening of the dermoepidermal junction, nuclear pyknosis, apoptotic bodies, irregular basal cell size and hypertrophy, fibroblast proliferation, and significant inflammatory infiltration [26,27]. UV radiation also induces collagen fiber fragmentation, denaturation, and subsequent reactive hyperplasia, leading to dermal thickening [28,29]. Additionally, the UV-mediated depletion of type I and III collagen is a major contributor to photoaging [27]. Previous studies have confirmed a marked reduction in type I collagen in photoaged mouse skin compared to normal controls [30].

In this study, dorsal skin sections from mice were subjected to HE and Masson staining to evaluate epidermal–dermal structural changes and collagen fiber morphology, while immunohistochemical staining was employed to assess differences in COL1A1-positive area ratios among the groups. The experimental results were largely consistent with previous findings. H&E staining revealed that the model group exhibited epidermal thickening, dermal fiber fragmentation, intercellular edema, and inflammatory infiltration. Both EGCG and EGCG-NPs ameliorated the structural alterations induced by photoaging, with EGCG-NPs demonstrating more pronounced improvements, which aligned with the epidermal thickness measurements obtained using the Dermalab Combo4 ultrasonic probe. Masson staining further indicated that EGCG-NPs more effectively restored the organization of dermal collagen fibers disrupted by photodamage and increased collagen content. Similarly, immunohistochemical analysis of COL1A1 showed that the DPNP group exhibited a higher COL1A1-positive area ratio compared to both the model and EGCG groups, consistent with the trends in collagen density data measured by the Dermalab Combo4 ultrasonic probe. Collectively, based on macroscopic observations, skin barrier function assessments, and histopathological analyses, it is concluded that both EGCG and EGCG-NPs can mitigate UV-induced skin damage, with EGCG-NPs demonstrating significantly superior therapeutic efficacy.

The mechanisms underlying photoaging are complex, and studies have shown that skin photoaging is primarily attributed to the UV-radiation-induced excessive production of reactive oxygen species (ROS) and free radicals, which cause damage to biomacromolecules within mitochondria, nuclei, and the cellular matrix. This damage triggers a cascade of responses, including acute and chronic inflammation, immune suppression, cell cycle alterations, and oxidative stress, ultimately leading to increased degradation and reduced synthesis of elastic and collagen fibers, thereby altering skin structure and resulting in photoaging [21,31,32,33,34]. A hallmark of skin photoaging is the degradation of the extracellular matrix (ECM). Research has demonstrated that fibroblasts in photoaged skin produce elevated levels of matrix metalloproteinases (MMPs), which degrade various protein components of the ECM [35]. Specifically, MMP-1 induces collagen fiber cleavage, contributing to skin roughness and laxity [36]. Studies have also indicated that inhibiting the expression and secretion of MMP-1 and MMP-3, as well as suppressing the expression and secretion of pro-inflammatory cytokines such as IL-6 and IL-1β, can promote collagen synthesis and mitigate photoaging damage [37]. Furthermore, pharmacological interventions have been shown to significantly reduce malondialdehyde (MDA) levels in skin tissue while enhancing the activity of antioxidant enzymes such as glutathione peroxidase (GSH-Px) and superoxide dismutase (SOD). Concurrently, these interventions markedly inhibit the secretion of inflammatory cytokines (IL-1β, IL-6) and MMPs (MMP-1, MMP-3), thereby achieving anti-photoaging effects [38].

In previous studies on skin photoaging, UV irradiation has been demonstrated to induce oxidative stress responses, leading to the depletion of endogenous antioxidant enzymes and the increased production of malondialdehyde (MDA), a toxic degradation product derived from the peroxidation of polyunsaturated fatty acids cross-linked with lipoproteins [39]. Consistent with these established findings, our experimental model group exhibited decreased activities of antioxidant enzymes (SOD, CAT, and GSH-Px) along with elevated MDA levels. Both EGCG and EGCG-NPs demonstrated significant efficacy in ameliorating UV-induced oxidative damage, with EGCG-NPs showing superior therapeutic effects. Notably, EGCG-NP treatment resulted in a particularly remarkable enhancement (*p* < 0.01) of GSH-Px activity, which functions as a free-radical-scavenging enzyme capable of eliminating free radicals and their derivatives while reducing lipid peroxidation products.

Research has demonstrated that IL-1β and IL-6 can stimulate the proliferation of keratinocytes and fibroblasts, as well as enhancing the chemotactic activity of keratinocytes [40]. Studies utilizing EGCG for treatment have shown that EGCG significantly suppresses the upregulation of IL-1β and IL-6 expression in aged cells, exhibiting potent anti-inflammatory and anti-aging properties [41]. In this experiment, UV radiation led to a marked increase in IL-1β and IL-6 levels in the model group compared to the Normal group. Both the EGCG group and the DPNP group inhibited the elevation of IL-1β, with the DPNP group demonstrating more pronounced suppression. However, while EGCG did not significantly inhibit IL-6 compared to the model group, EGCG-NPs exhibited a highly significant inhibitory effect on IL-6 (*p* < 0.01). This enhanced efficacy of EGCG-NPs may be attributed to the nanostructure improving the stability and bioavailability of EGCG, thereby amplifying its anti-inflammatory effects.

Extensive research has established that matrix metalloproteinases (MMPs) serve as the primary mediators responsible for the degradation of extracellular matrix (ECM) components, including collagen, fibronectin, elastin, and proteoglycans [42]. In the present study, ELISA analysis of MMP-1 and MMP-3 revealed that both EGCG and EGCG-NPs significantly downregulated the expression of these proteolytic enzymes compared to the model group, with EGCG-NPs demonstrating superior inhibitory efficacy. Notably, EGCG-NP treatment resulted in a more pronounced reduction in MMP-1 expression (*p* < 0.01) relative to conventional EGCG, a finding that aligns with the observed trends in COL1A1 immunohistochemical staining results.

In this study, we further conducted ELISA analysis of the cytokine NF-κB, with our results being consistent with established theoretical mechanisms of skin photoaging and previous research findings. Current evidence indicates that NF-κB serves as a pivotal regulator in generating the senescence-associated secretory phenotype (SASP) and represents a crucial pathway mediating ROS-induced inflammatory responses. Following UV irradiation, activated NF-κB subunits translocate into the nucleus, triggering the upregulation of pro-inflammatory cytokines while simultaneously inducing the elevated expression of MMPs [42,43,44]. Notably, research has demonstrated that EGCG can effectively mitigate inflammatory responses by suppressing NF-κB activation and subsequent IL-6 secretion, while concurrently downregulating MMP expression to prevent collagen degradation, thereby exerting anti-photoaging effects [36,45,46]. Our experimental data revealed that, compared to the Normal group, UV exposure significantly increased NF-κB expression in the model group. Both EGCG and EGCG-NP treatments markedly attenuated this UV-induced NF-κB upregulation, with EGCG-NPs exhibiting superior inhibitory efficacy (*p* < 0.01).

The experimental results demonstrate that EGCG exhibits significant ameliorative effects on cutaneous photoaging. Notably, the EGCG-NPs demonstrate enhanced therapeutic efficacy through multiple mechanisms, including effectively attenuating the UV-radiation-induced suppression of antioxidant enzyme activities, significantly inhibiting UV-activated inflammatory cytokines, downregulating matrix metalloproteinase expression, and suppressing NF-κB pathway activation. These superior pharmacological effects substantiate that the nanostructure significantly enhances the bioavailability of EGCG in vivo, thereby optimizing its anti-photoaging potential. However, the current investigation into the mechanisms of EGCG-NPs remains somewhat limited, primarily focusing on their antioxidative and anti-inflammatory properties. In addition to the antioxidant enzymes investigated in this study, research has demonstrated that Nrf2 plays a critical role in regulating antioxidant responses in various tissues, including the skin. As a key transcription factor in redox processes, Nrf2 has been shown to mitigate photoaging, promote wound healing, and prevent hyperpigmentation. It also protects against UV-induced skin cancer by maintaining skin structural integrity and regulating skin homeostasis [47,48]. The mechanisms by which Nrf2 ameliorates skin photoaging are multifaceted, typically involving antioxidant activity, cellular damage repair, DNA repair, and anti-inflammatory effects [49]. Experimental studies have observed that Nrf2 can reduce melanogenesis in melanocytes by downregulating signaling pathways (e.g., the cAMP/CREB/MITF pathway) through the inhibition of paracrine factors (e.g., α-MSH) derived from keratinocytes [50]. Furthermore, Nrf2’s anti-photoaging effects are closely linked to inflammatory responses. For instance, the UV-radiation-induced downregulation of Nrf2 signaling has been shown to upregulate pro-inflammatory factors in keratinocytes, including tumor necrosis factor-α (TNF-α), cyclooxygenase-2 (COX-2), interleukin-6 (IL-6), interleukin-1β (IL-1β), and interleukin-8 (IL-8) [51]. Multiple signaling pathways, such as NF-κB, MAPK, and JAK-STAT, are involved in these processes. Notably, previous studies have demonstrated the interaction between Nrf2 and NF-κB, revealing that Nrf2 exerts anti-inflammatory effects by suppressing oxidative-stress-induced NF-κB activation and inhibiting the proteasomal degradation of IκB-α (an NF-κB inhibitor) [52,53]. In summary, Nrf2 is a pivotal regulator of redox reactions and is associated with multiple pathways, including anti-inflammatory mechanisms, making it essential for combating skin photoaging. However, this study did not include Nrf2 detection, and thus cannot further validate the findings of previous research. Future studies could incorporate relevant experiments to explore the relationships between these factors in greater detail. Also, to advance this research domain, future studies should incorporate a more comprehensive mechanistic exploration, including the characterization of the nanocarrier’s intrinsic physicochemical properties and biological functions, the elucidation of the intestinal transport mechanisms of EGCG-NPs, and the evaluation of their potential modulatory effects on gut microbiota. Such multifaceted investigations would not only expand the identified therapeutic targets of EGCG-NPs but also provide substantial experimental evidence to facilitate the refinement of nanocarrier systems for optimized drug delivery.

## 5. Conclusions

In the present study, both EGCG and EGCG-NPs were observed to exert significant ameliorative effects on UV-induced photoaging in mice. Notably, compared to conventional EGCG, EGCG-NPs demonstrated superior efficacy in improving multiple parameters of cutaneous photoaging, including the macroscopic appearance of photoaged skin, skin barrier function, collagen content, and UV-induced alterations in antioxidant and anti-inflammatory markers. These findings strongly suggest that the developed EGCG-NP formulation effectively enhances the bioavailability of EGCG in vivo, thereby potentiating its photoprotective effects in murine models. The observed therapeutic advantages of EGCG-NPs underscore their potential as promising candidates for further investigation and development in the field of dermatological research.

## Figures and Tables

**Figure 1 foods-14-02150-f001:**
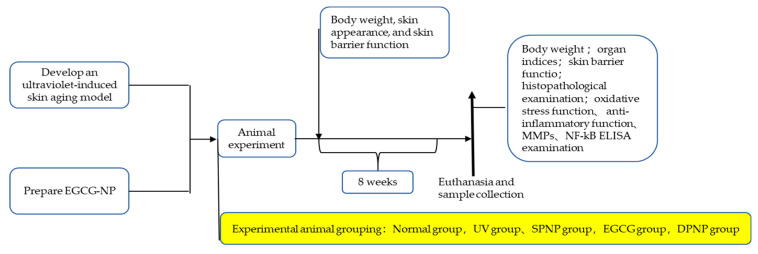
Experimental schedule.

**Figure 2 foods-14-02150-f002:**
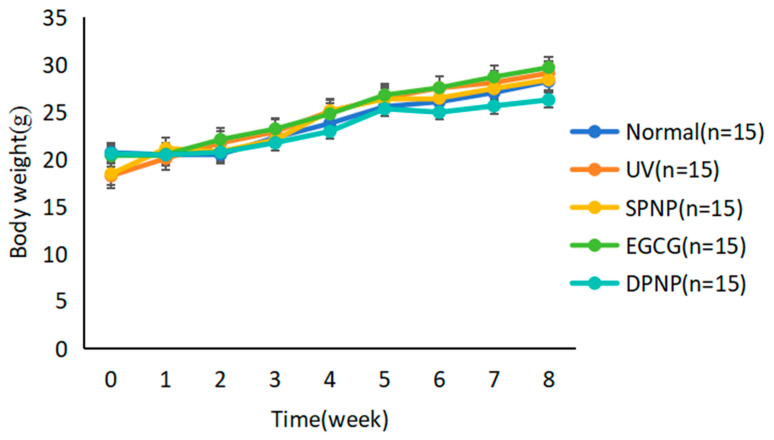
The changes in body weight measured during the 8 weeks of the experiment (mean ± SD).

**Figure 3 foods-14-02150-f003:**
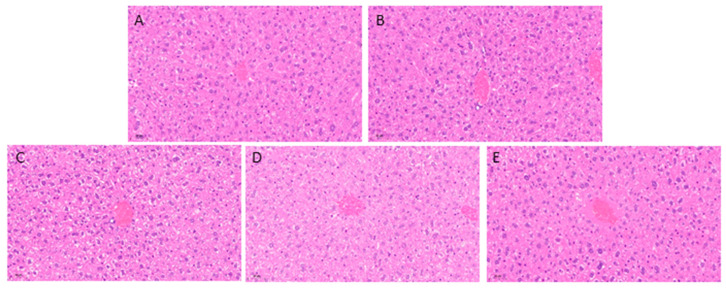
Changes in hepatic histopathology. (**A**) Normal group. (**B**) UV group. (**C**) SPNP group. (**D**) EGCG group. (**E**) DPNP group. Scale bar: 50 μm. (n = 15 per group).

**Figure 4 foods-14-02150-f004:**
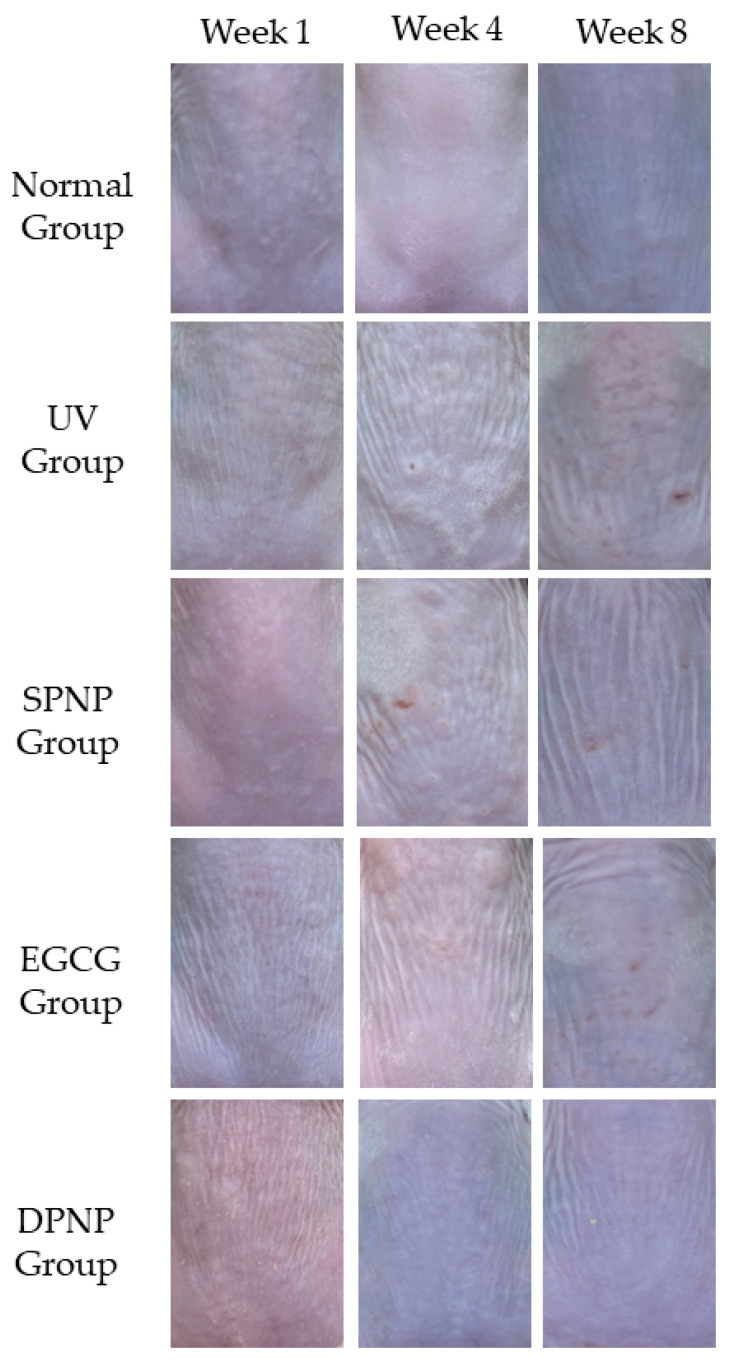
Dorsal skin condition of mice in each group at Weeks 1, 4, and 8 (n = 15 per group).

**Figure 5 foods-14-02150-f005:**
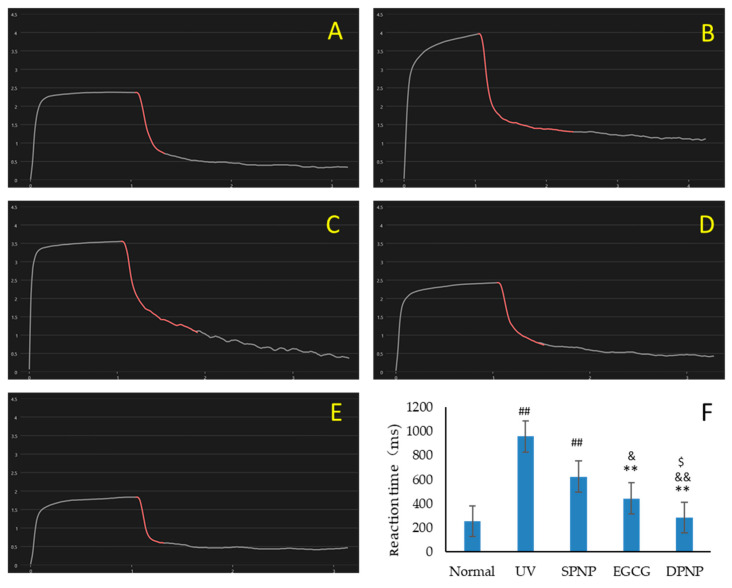
Skin elasticity analysis. (**A**) Normal group. (**B**) UV group. (**C**) SPNP group. (**D**) EGCG group. (**E**) DPNP group. (**F**) Comparative analysis of elastic recoil time across experimental groups. Each bar represents mean ± SD (n = 15 per group).The red segment of the elastic curve represents weakened skin rebound capacity, indicating the onset of skin laxity, before ultimately returning to the skin’s normal steady state. ^##^
*p* < 0.01 compared to Normal group. ** *p* < 0.01 compared to UV group. ^&^
*p* < 0.05 compared to SPNP group. ^&&^
*p* < 0.01 compared to SPNP group. ^$^
*p* < 0.05 compared to EGCG group.

**Figure 6 foods-14-02150-f006:**
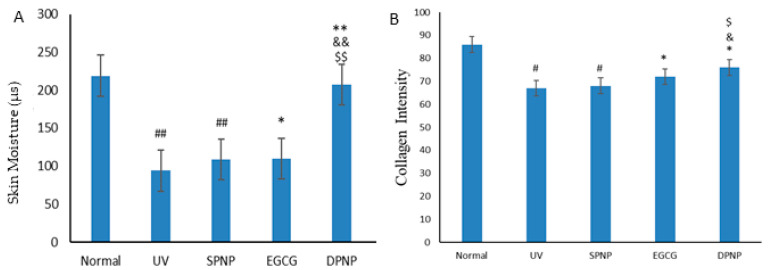
Cutaneous hydration status and collagen density assessment. (**A**) Comparative analysis of cutaneous hydration status. (**B**) Comparative analysis of collagen density assessment. Each bar represents mean ± SD (n = 15 per group). ^#^
*p* < 0.05 compared to Normal group. ^##^
*p* < 0.01 compared to Normal group. * *p* < 0.05 compared to UV group. ** *p* < 0.01 compared to UV group. ^&^
*p* < 0.05 compared to SPNP group. ^&&^
*p* < 0.01 compared to SPNP group. ^$^
*p* < 0.05 compared to EGCG group. ^$$^
*p* < 0.01 compared to EGCG group.

**Figure 7 foods-14-02150-f007:**
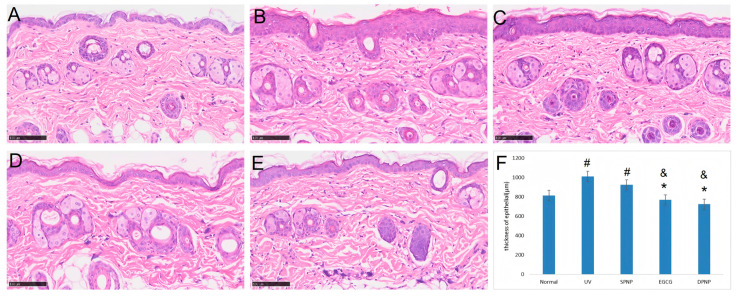
Histopathological evaluation of skin by hematoxylin–eosin (H&E) staining. (**A**) Normal group. (**B**) UV group. (**C**) SPNP group. (**D**) EGCG group. (**E**) DPNP group. (**F**) Comparative analysis of epidermal thickness among experimental groups. Each bar represents mean ± SD (n = 15 per group). Scale bar: 100 μm. ^#^
*p* < 0.05 compared to Normal group. * *p* < 0.05 compared to UV group. ^&^
*p* < 0.05 compared to SPNP group.

**Figure 8 foods-14-02150-f008:**
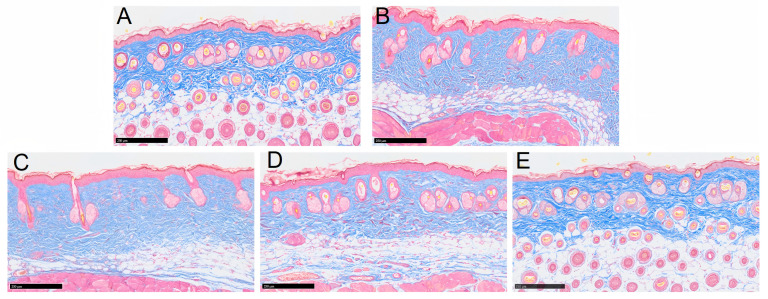
Histopathological evaluation of skin by Masson staining. (**A**) Normal group. (**B**) UV group. (**C**) SPNP group. (**D**) EGCG group. (**E**) DPNP group. (n = 15 per group). Scale bar: 250 μm.

**Figure 9 foods-14-02150-f009:**
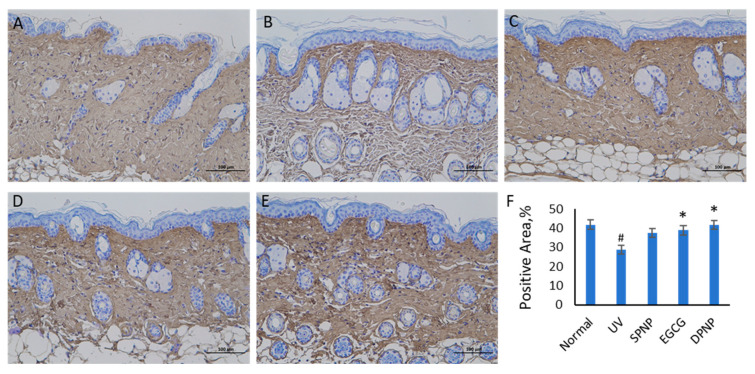
COL1A1 spatial distribution by immunohistochemical analysis. (**A**) Normal group. (**B**) UV group. (**C**) SPNP group. (**D**) EGCG group. (**E**) DPNP group. (**F**) Comparative analysis of percentage of positive staining area for COL1A1. Each bar represents mean ± SD (n = 3 per group). Scale bar: 100 μm. ^#^
*p* < 0.05 compared to Normal group. * *p* < 0.05 compared to UV group.

**Figure 10 foods-14-02150-f010:**
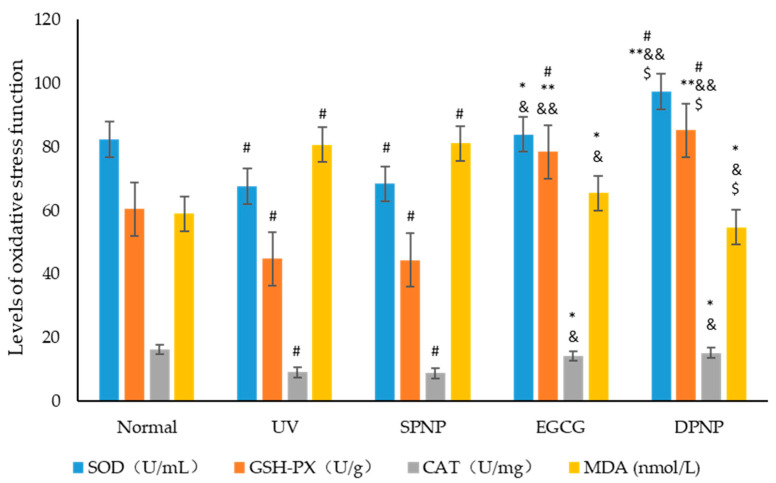
Levels of SOD, CAT, GSH-Px, and MDA in each experimental group (Mean ± SD). Each bar represents mean ± SD (n = 15 per group). ^#^
*p* < 0.05 compared to Normal group. * *p* < 0.05 compared to UV group. ** *p* < 0.01 compared to UV group. ^&^
*p* < 0.05 compared to SPNP group. ^&&^
*p* < 0.01 compared to SPNP group. ^$^
*p* < 0.05 compared to EGCG group.

**Figure 11 foods-14-02150-f011:**
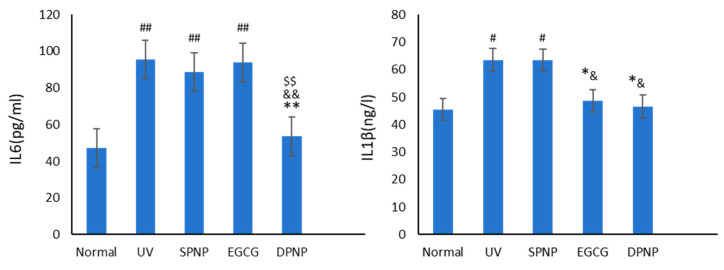
Levels of IL6 and IL1β in each experimental group. Each bar represents mean ± SD (n = 15 per group). ^#^
*p* < 0.05 compared to Normal group. ^##^
*p* < 0.01 compared to Normal group. * *p* < 0.05 compared to UV group. ** *p* < 0.01 compared to UV group. ^&^
*p* < 0.05 compared to SPNP group. ^&&^
*p* < 0.01 compared to SPNP group. ^$$^
*p* < 0.01 compared to EGCG group.

**Figure 12 foods-14-02150-f012:**
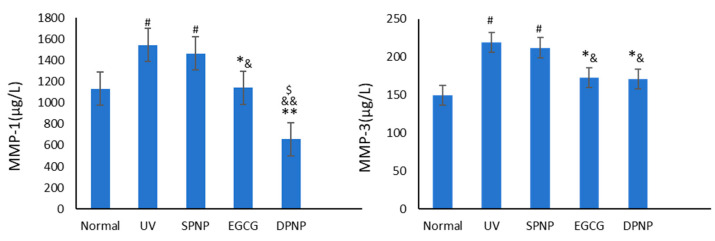
Levels of MMP-1 and MMP-3 in each experimental group. Each bar represents mean ± SD (n = 15 per group). ^#^
*p* < 0.05 compared to Normal group. * *p* < 0.05 compared to UV group. ** *p* < 0.01 compared to UV group. ^&^
*p* < 0.05 compared to SPNP group. ^&&^
*p* < 0.01 compared to SPNP group. ^$^
*p* < 0.05 compared to EGCG group.

**Figure 13 foods-14-02150-f013:**
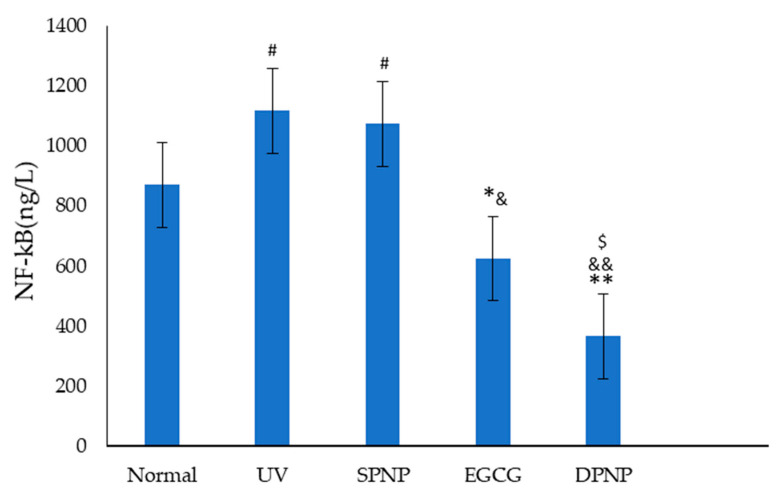
Levels of NF-κB in each experimental group. Each bar represents mean ± SD (n = 15 per group). ^#^
*p* < 0.05 compared to Normal group. * *p* < 0.05 compared to UV group. ** *p* < 0.01 compared to UV group. ^&^
*p* < 0.05 compared to SPNP group. ^&&^
*p* < 0.01 compared to SPNP group. ^$^
*p* < 0.05 compared to EGCG group.

**Table 1 foods-14-02150-t001:** Animal grouping and experimental treatments.

Group	Number	Intervention Protocol
Normal	15	received no treatment for 8 weeks
UV	15	UV irradiation for 5 min once daily for 8 weeks
SPNP	15	UV irradiation for 5 min + oral gavage with 0.1 mL/10 g blank nanoparticle solution once daily for 8 weeks
EGCG	15	UV irradiation for 5 min + oral gavage with 0.1 mL/10 g EGCG solution (1.1 mg/g.bw) once daily for 8 weeks
DPNP	15	UV irradiation for 5 min + oral gavage with 0.11 mL/10 g EGCG nanoparticle solution (1 mg/mL) once daily for 8 weeks

**Table 2 foods-14-02150-t002:** Organ indices of mice in each group (n = 15 per group, mean ± SD).

Group	Liver Index	Spleen Index	Kidney Index	Heart Index
Normal	5.4 ± 0.24	0.4 ± 0.06	1.8 ± 0.13	0.7 ± 0.13
UV	5.4 ± 0.23	0.4 ± 0.05	1.9 ± 0.12	0.6 ± 0.08
SPNP	5.4 ± 0.23	0.4 ± 0.08	1.8 ± 0.10	0.6 ± 0.15
EGCG	5.5 ± 0.41	0.4 ± 0.06	1.9 ± 0.09	0.7 ± 0.11
DPNP	5.6 ± 0.41	0.6 ± 0.10	1.9 ± 0.12	0.8 ± 0.12

## Data Availability

No new data were created or analyzed in this study. Data sharing is not applicable to this article.

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
