# Peer review of "An Elucidation of the Anti-Photoaging Efficacy and Molecular Mechanisms of Epigallocatechin Gallate Nanoparticles in a Balb/c Murine Model"

_foods, 2025, doi:10.3390/foods14132150_

Round 1

Reviewer 1 Report

Comments and Suggestions for Authors

This manuscript addresses a relevant and timely topic in the field of dermatological therapeutics by evaluating the anti-photoaging potential of EGCG-loaded zein-chitosan nanoparticles. The study is well-structured, and the authors provide a comprehensive experimental design that includes histological, biochemical, and molecular assessments. The findings are generally consistent and support the proposed hypothesis that nanoparticle-based delivery enhances the efficacy of EGCG against UV-induced skin damage. However, there are several issues related to figure clarity, formatting, and methodological descriptions that need to be addressed to improve the manuscript’s overall quality and reproducibility. My detailed comments are provided below.

General

  • Please carefully review the manuscript for spacing issues and typographical errors throughout the text.

Abstract Section

  • Line 24: Please standardize the abbreviation format for "Hematoxylin and Eosin" as "H&E" throughout the manuscript.
  • Line 35: Change “ef-fects” to “effects.”

Materials and Methods Section

  • Please consider including a schematic or figure that shows the overall experimental schedule.
  • Line 134: Replace “issue” with “tissue.”
  • Was the intestine collected in this study? If so, please clarify whether there was any change in its organ index.

Results Section

  • Please check the spacing between the subheading numbers and the subheading titles.
  • Unify the font type and size across all figure graphs for consistency.
  • Ensure the figure legend format is consistent—either all short or all detailed.
  • Line 185: The description of “mental state” and “slow movement” is vague. Please clarify how these were evaluated and whether any quantitative measurements are available.
  • Figure 2: A scale bar is missing in the image. Please add a clearly visible scale bar.
  • Figure 3: Please revise the notation “n=15” to match the journal’s formatting standard, e.g., “n = 15 per group.”
  • Figure 4: Error bars are not visible. Please recheck the figure and replace it with a clearer version.
  • Figure 9: Please re-evaluate the statistical comparison between the EGCG group and the normal group. Is it accurate that there is no significance between them? Also, align the significance indicators consistently across the graph.
  • Figure 11: In the last line of the figure legend, the phrase “3.1 Subsection” appears to be irrelevant. Please remove it if it is not applicable.

Discussion Section

  • Line 450, 459: Please italicize the letter “P” when referring to p-values (e.g., P < 0.05).

Additional Comment on Scientific Rationale

  • In both the Introduction and Discussion, the authors mention that nanoparticle formulation improves intestinal retention time, leading to enhanced bioavailability. However, the cited references do not include in vivo results that support this claim regarding intestinal retention time.
  • Furthermore, in the current 8-week oral administration of EGCG-NP, were there any assessments conducted on the intestine to support this claim? If not, this limitation should be acknowledged and discussed.

Reviewer 2 Report

Comments and Suggestions for Authors

The manuscript entitled « Elucidation of the Anti-Photoaging Efficacy and Molecular Mechanisms of Epigallocatechin Gallate Nanoparticles in a Balb/c Murine Model » has dealt with investigating the effect of both EGCG and EGCG-NP on UV-induced photoaging in mice.

Abstract :

Comment 1: Several words in the manuscript are incorrectly hyphenated, for example, re-duced. I recommend reviewing the entire text.

Introduction :

Comment 2: Lines 51-58: I recommend developing this part and mentioning the composition of both anti-photoaging agents (chemical and physical) as well as their limitations.

Materials and Methods

Comment 3: Animal Care and Skin Aging Model: Add a recap table of the animal treatment.

Comment 4: Histological Examination of Liver and Skin Tissues: Mention the thickness of the paraffin sections.

Comment 5: Determination of Oxidative Stress, Inflammatory and Matrix Metalloproteinases (MMPs) Markers in Skin Tissue: This title includes 3 different markers, making it unclear. I recommend dividing it into 3 separate titles with one title dedicated to each marker. In addition, develop each protocol providing the objective, the principle and the detailed steps of the procedure.  

Comment 6: I recommend including the preparation of the homogenate.

Comment 7: Measurement of NF-κB Signaling Pathway Activity: please provide the link to the manufacturer's protocol.

Results :

Comment 8: Mention in the Materials and Methods section that the mice were weighed and specify how many times during the treatment.

Comment 9: Increase the resolution of all figures.

Comment 10: Figure 1, 7 and 8 : Please draw the scale bar on each image.

Comment 11: Figure 3 : Add a detailed legend.

Comment 12: Table 2: I suggest representing the results in the form of histograms for more clarity and better visualization.

Discussion :

Comment 13: When discussing oxidative stress, the Nrf2 signalling pathway plays a key regulatory role. Develop the Nrf2 signalling pathway explaining how UV and treatment change the activity of the three markers of oxidative stress (SOD, CAT and GPx) since Nrf2 is a transcription factor that regulates the expression of various antioxidant genes.

Comment 14: When discussing inflammation, the NF-κB signalling pathway plays a key regulatory role. Therefore, please discuss the results in the context of this pathway. Develop the NF-κB signalling pathway explaining how UV and treatment change the inflammatory parameters.  

Round 2

Reviewer 2 Report

Comments and Suggestions for Authors

I have carefully reviewed the revised manuscript and the authors' detailed responses to my previous comments. I am pleased to confirm that the authors have adequately and satisfactorily addressed all the concerns I raised in my initial review. The revisions have significantly improved the scientific rigor, clarity, and overall quality of the manuscript, and I believe it is now suitable for publication in its current form. I strongly recommend that this paper be accepted for publication.